# Applying an osteopathic intervention to improve mild to moderate mental health symptoms: a mixed-methods feasibility study protocol

Josh Hope-Bell,[1,2] Jerry Draper-Rodi [ID],[3,4] Darren J Edwards [ID] [1]

[1]Department of Public Health, Swansea University, Swansea, UK
[2]Division of Psychological Medicine and Clinical Neurosciences, School of Medicine, Cardiff University, Cardiff, UK
[3]University College of Osteopathy, London, UK
[4]National Council for Osteopathic Research, London, UK

**Correspondence to**
Dr Darren J Edwards;
D.J.Edwards@swansea.ac.uk

## ABSTRACT

**Introduction** Mental health services are stretched in the UK and are in need of support. One approach that could improve mental health symptoms is osteopathy. Research suggests that osteopathy influences psychophysiological factors, which could lead to improvements in mental health. The first objective of this protocol is to investigate the feasibility and acceptability of four osteopathic interventions. A secondary aim is to evaluate the interventions' effectiveness for improving psychophysiological and mental health outcomes.

**Methods and analysis** This study will be an explanatory mixed-methods design. Participants will be 30 adults who have mild to moderate mental health symptoms and not experiencing any issues with pain. The feasibility and acceptability of the interventions will be the primary outcomes. Secondary outcomes will be physiological measures including heart rate variability, interoceptive accuracy and blood pressure. Psychological outcomes, collected preintervention and postintervention, will also be measured by five standardised questionnaires, which include: (1) the Depression, Anxiety and Stress Scale (DASS); (2) the International Positive and Negative Affect Schedule-Short-Form; (3) Acceptance and Action Questionnaire-II; (4) the Self as Context Scale and (5) and the Multidimensional Assessment of Interoceptive Awareness Version 2. Participants will be randomised to one of four intervention groups and receive a single intervention treatment session. These intervention groups are: (1) high-velocity and articulation techniques, (2) soft-tissue massage, (3) craniosacral techniques, and (4) a combination of these three approaches. Mixed design two (preintervention and postintervention) by the four interventions analysis of covariance models will be used to analyse the quantitative data for each quantitative measure. Participants will also be interviewed about their experiences of the study and interventions and a thematic analysis will be used to analyse this qualitative data. This will aid the assessment of the feasibility and acceptability of the study design.

**Ethics and dissemination** The protocol for this feasibility study has received ethical approval from the Department of Psychology Ethics Committee at Swansea University, ethical review reference number: 2022-5603-4810. Feasibility results from this protocol will be published in a peer review journal and presented at both national and international conferences.

## STRENGTHS AND LIMITATIONS OF THIS STUDY

⇒ This study will investigate the utility of osteopathic techniques for improving mental health, in the absence of any existing pain.
⇒ The techniques being compared are based on previous literature and evidence.
⇒ Due to this being a feasibility study, only a small number of participants are being recruited which will lead to low statistical power of the results.
⇒ As the study is focused on comparing four interventions in a feasibility setting, there will be no control group comparison.

**Discussion** This study will assess the feasibility and acceptability of conducting osteopathic interventions for improving mental health outcomes. The results from this will help to inform the development of a future randomised controlled trial. The study will also produce original data which could provide preliminary evidence of whether osteopathic approaches are of benefit to individual's mental health in the form of effect sizes, even if they are pain-free.

**Trial registration number** NCT05674071.

## INTRODUCTION
### Background and rationale
In the UK, mental health problems such as anxiety and depression are an increasing burden within society. Recent estimates suggest that one in six people in the UK experience symptoms of depression or anxiety in any given week.[1] For the individual, poor mental health can bring about problematic coping behaviours such as substance abuse and self-harm, leading to poor social relationships and in the worst cases; suicide.[2] Mental health problems are commonly treated through psychotherapeutic means such as cognitive–behavioural therapy, acceptance and commitment therapy, as well as relaxation techniques such as mindfulness practice and yoga.[3] In addition, pharmacological solutions such as antidepressants and beta-blockers are

used in treatment. These approaches have demonstrated effectiveness in many cases though they treat the symptoms and not the underlying causes.[4 5]

With such a high demand being placed on the health services, such as these traditional forms of care, it can be difficult for many to receive treatment.[6 7] It may, therefore, be important and helpful to consider innovative approaches that could support the demand for mental health services.[8] Recently, it has been suggested that osteopathic interventions could be one such approach to support mental health services.[9 10]

Osteopathy is an approach to healthcare that uses manual techniques to diagnose and treat patients.[11] Osteopathy is an Allied Health Profession in England and osteopaths in the UK are regulated by statute.[12] An osteopathic approach is patient-centred and focused on the patient's health rather than disease-centred. The practices are evidence-informed and scientific rigour forms an important part of treating patients and managing cases.[13] Osteopaths use manual contact to identify and evaluate movement in all structural and functional aspects of the patient, identifying alterations of function and movement that impede health and addressing these. Osteopaths use a variety of techniques to manipulate joints, muscles and tissue. All of the techniques used have an effect on the interplay between the nervous and musculoskeletal systems.[14–16] Specific techniques include myofascial release, lymphatic drainage, high-velocity, low amplitude, articulatory techniques and muscle energy techniques.

The rationale for linking mental health with physiological mechanisms comes as previous studies have examined how osteopathy may influence psychophysiological factors. A number of these have examined the influence of osteopathic manipulative therapy (OMT) on heart rate variability (HRV), which is considered a potentially important indicator of physical and psychological well-being.[17 18] Cerritelli et al[19] found that two sessions of OMT significantly increased HRV in healthy adults, relative to a sham control group. Similarly, Arienti et al[20] found that applying a single session of fourth ventricle compression (CV4) significantly increased HRV, compared with a placebo intervention. So, there seems to be some clear evidence that influencing physiological mechanisms through osteopathy is highly relevant for mental health improvement. Indeed, studies have found that osteopathic interventions have led to improvements in mental health outcomes such as anxiety,[21] depression[22] and stress.[23] However, few studies have sought to examine the effects of osteopathy on both physiological and psychological outcomes.

This feasibility protocol (that has followed the Standard Protocol Items: Recommendations for Intervention Trials (SPIRIT) guidelines, see online supplemental material 1)[24] will, therefore, explore both the potential psychological and psychophysiological changes that occur after osteopathic treatment, in a small group of individuals who suffer from mild to moderate forms of anxiety, stress and depression.

## Choice of comparators

This study will be comparing four osteopathic interventions: (1) articulation and high-velocity thrust (AHVT) techniques, (2) soft-tissue massage (STM), (3) craniosacral techniques (CST) and (4) a combination of all three techniques.

The choice of techniques in this study has been informed by a systematic review and meta-analysis by the authors into the impact of osteopathic interventions on psychophysiological factors (This systematic review has been preregistered on Open Science Framework (OSF) and the protocol is available via this link: https://osf.io/jrtpx/?view_only=63ffa916b76c4b95b4233d3c d812f12d). This review included randomised controlled trials (RCTs) of manual interventions and their effects on factors including mental health outcomes and physiological indicators such as HRV and interoception. Articulation techniques were found to improve psychological outcomes,[25 26] as well as autonomic nervous system indicators such as HRV and interoception.[19] Similarly, interventions utilising HVT improved interoceptive accuracy (IAc) and led to greater activation of brain areas associated with the interoceptive pathways.[27] The studies that suggested articulation could improve psychological outcomes were conducted with chronic pain patients and the study on HVT with healthy participants. It will, therefore, be useful to understand whether articulation techniques and HVT could have positive psychological effects in the absence of pain and in the presence of mild mental health symptoms.

Next, STM techniques were chosen. Studies show that this approach has several positive psychological impacts for individuals with chronic pain[22] and pain-free patients who have mental health diagnoses.[21 28] Massage therapy has also been shown to have a preventative effect for general stress and well-being.[23] Lastly, massage techniques have been shown to induce autonomic relaxation in healthy participants by increasing HRV.[29]

The third intervention will use CST. Three studies suggested that this approach induces autonomic relaxation by increasing HRV. One of these studies was conducted with patients with chronic pain[30] and two were carried out with healthy participants.[20 31] It will, therefore, be useful to see whether any potential autonomic changes from CST translate to psychological benefits. It will also be useful to examine the potential utility of these techniques with participants who have mild mental health symptoms.

The body areas that each intervention will focus on have also been informed by the aforementioned literature. The interventions will operate on a standardised protocol whereby body areas will be worked on in order by the practitioner. The body areas focused on and the order they will be worked on will be described further in the specific procedures of each intervention.

## Objectives

This study aims to investigate the feasibility and acceptability of four osteopathic interventions for adults with mild to moderate mental health symptomatology. A secondary aim is to evaluate the influence of these four interventions on physiological factors including HRV and interoception. The study will aim to evaluate their effectiveness in improving psychological outcomes including depression, stress, anxiety, negative affect (NA) and psychological flexibility. It is first hypothesised that the interventions will be feasible and acceptable to participants. It is also hypothesised that the interventions will induce psychophysiological relaxation by significantly increasing HRV and improving IAc. No specific predictions are made about blood pressure (BP) as data on this is only being collected to check that we are within safe BP bounds, that is, for participant safety. A final hypothesis is that the four interventions will lead to similar improvements in depression, anxiety, stress and NA.

## Trial design

This is a feasibility study, which will use an explanatory sequential mixed-methods approach. In this approach, the quantitative aspect forms the first part of the study, followed by a qualitative aspect to help provide further explanation and depth.[32] For the quantitative aspect, the study will use a parallel, randomised design with an equal proportion of participants allocated to each of the four conditions. The qualitative aspect will be completed by interviewing the participants of the intervention and the practitioner delivering them.

## METHODS: PARTICIPANTS, INTERVENTIONS AND OUTCOMES
### Study setting

The study will take place at Swansea University in South Wales, UK with participants being recruited from both the student population at the university and the general public. The interventions will only take place in one location and country: Wales, UK. The study began on 20 December 2022, and the study will be completed by 1 August 2023.

### Eligibility criteria

Eligibility criteria will include being over 18 years of age, experiencing mild to moderate symptoms of depression, stress, or anxiety, and being able to read, write and speak English. Prospective participants will be excluded if they are experiencing acute or chronic pain, and/or if they have no psychological symptoms or more severe mental health issues. The rationale for excluding participants with pain is that it may present a confounding variable. That is, if the osteopathic intervention alleviates any pain the participants are experiencing, this may lead to improvements in psychological symptoms. It would, therefore, not be clear whether osteopathy has a more direct influence on mental health outcomes. Screening

for mild to moderate psychological symptoms will be conducted using the DASS.

## Interventions

Participants will receive one of four interventions based on osteopathic techniques. All four interventions will consist of a single session lasting approximately 30 min. The interventions are being delivered by two male osteopaths, one with 17 years of practice experience and one with 3 years of practice experience. The interventions will be as follows: (1) articulation and HVT techniques, (2) STM, (3) CST and (4) a combination of all three techniques. A summary of the intervention protocols can be found in table 1.

### Articulation and high-velocity techniques

The AHVT intervention will begin with an examination of the participant to search for somatic dysfunction.[33] The AHVT intervention will primarily be targeting all areas of the participant's spine. That is, the cervical, thoracic and lumbar areas, and also the sacroiliac joints. The practitioner will first observe the participant while standing, then will observe active range of movements with the participant in standing and/or sitting positions. Then the practitioner will continue their examination searching first by light and then deeper palpation for signs associated with somatic dysfunction with the participant sitting down or lying prone or supine. This segment of the intervention will be allocated approximately 10 min.

If areas of the spine are found to have somatic dysfunction, then AHVT techniques will be applied to these areas. If no areas of somatic dysfunction are identified in the aforementioned spinal areas, then the practitioner will first focus on applying AHVT techniques to the thoracic spine and rib cage areas, followed by articulation techniques such as hip extension. The application of techniques will be allocated approximately 20 min.

### STM techniques

The STM intervention will be a full-body massage. The participant will first be in the prone position and the practitioner will massage the upper, middle and lower areas of the back, the upper buttocks, then the hamstrings and calves. This will be approximately 15 min. The participant will then move into the supine position where they will receive massage on their neck, shoulders, pectoral muscles, arms, quadriceps and feet. This will also be allocated approximately 15 min. The literature suggests that slower techniques such as Swedish massage demonstrate effectiveness.[23 28] There is also evidence that focusing on the upper layers of the skin has psychological benefits.[22] These techniques will, therefore, be employed here.

### Craniosacral techniques

This intervention will use CST. This approach targets the cranial muscles and muscles around the central nervous system.[34] The CST intervention will begin with

**Table 1** Summary of the four intervention protocols and procedures

| | | Duration |
|---|---|---|
| For all | 1. 30 min appointment<br>2. Clinical findings, intervention, consent and adverse events (separate form to use if they do) recorded in participant individual form | |
| Articulation/HVT group | 1. Observation+AROM (standing or sitting)+clinical examination for SD (sitting, prone or supine)<br>2. Techniques:<br> 1. SD found: HVT to the area unless contraindicated (info on BP/HA)<br> 2. No SD found:<br> 1. HVT TSp and ribs<br> 2. Articulation of hips in extension | 1. 10 min<br>2. 20 min |
| Soft tissue group | Full body, slow and superficial<br>1. Prone:<br> 1. upper/mid/lower back<br> 2. upper buttocks<br> 3. hamstrings<br> 4. calves<br>2. Supine:<br> 1. Neck incl. suboccipital muscles<br> 2. Shoulders<br> 3. Pectoral muscles<br> 4. Arms<br> 5. Quadriceps<br> 6. Feet | 1. 15 min<br>2. 15 min |
| Cranial group | Looking for stiffness, asymmetry and tenderness on:<br>1. Sacrum<br>2. Head<br>Dysfunction found: myofascial release technique (10 min/area); if no dysfunction found: functional techniques applied to each area (10 min/area)<br>3. CV4 | 1. 10 min<br>2. 10 min<br>3. 10 min |
| Combined group | 1. Observation+AROM+clinical exam for SD<br>2. HVT TSp<br>3. Soft tissue upper and lower back prone<br>4. CV4 and suboccipital release | 1. 9 min<br>2. 7 min<br>3. 7 min<br>4. 7 min |

AROM, active range of movement; BP, blood pressure; CV4, Compression of the Fourth Ventricle technique; HA, headache; HVT, high velocity thrust techniques; SD, somatic dysfunction; TSp, thoracic spine.

an examination for somatic dysfunction such as stiffness, asymmetry or tenderness. The body areas focused on will be the soft tissue around the head and sacrum areas which are body areas commonly associated with CST. If areas of dysfunction are identified, then the practitioner will perform myofascial release. If no areas of dysfunction in these areas are identified, the practitioner will first focus on the sacral region and then move on to other areas associated with CST. For sacral and cranial areas, approximately 10 min will be allocated each for 20 min total. The intervention will conclude with fourth ventricle compression (CV4). This technique is performed on the occipital bone. CV4 will be allocated approximately 10 min of the intervention.

### Combination of techniques

This intervention will be a combination of all three techniques used in the other interventions (COMBO). The intervention will begin with an examination of the participant and checking active and passive range of movement. This examination will be allocated approximately

9 min. Using a combination of treatments, the intervention will consist of: (1) high-velocity techniques applied to the thoracic spine (approximately 7 min), (2) STM to the upper and lower back of the participant in prone (approximately 7 min), and (3) CV4 and suboccipital muscles release (approximately 7 min). This intervention will therefore last approximately 30 min.

### Modifications

In the interest of participant's safety, certain modifications may be made to the interventions if participants have body areas that are tender or if they present undiagnosed high BP (HBP). This is mostly relevant to the AVHT intervention and COMBO intervention which will have techniques that are of higher force. If a participant in the AVHT or COMBO interventions presents with HBP, neck pain or headaches during the intervention then the practitioner will not work on the cervical spine area and focus on the other spinal regions in the protocol. The justification is that HVT techniques may increase the risk of arterial damage in individuals with HBP.[35 36]

## Adherence

As the intervention only consists of one session, adherence is not necessarily applicable. Instead, a record will be kept of any participants who asked to end the intervention session early.

## Concomitant care

Participants will be asked at preintervention if they are receiving any drug treatment for mental health (eg, antidepressants), or psychotherapy (eg, cognitive behavioural therapy). Participants will not be excluded on this basis, but these will be factored into the main statistical analysis as covariates.

## Outcomes

The primary outcomes are the feasibility and acceptability outcomes (as described in Feasibility and Acceptability section), while the secondary outcomes are the psychological outcome measures (Mental health–Psychological flexibility) and the psychophysiological measures (Heart rate variability–Blood pressure).

## Feasibility

The feasibility of the recruitment process will be determined by the number of people who respond to the advertisements and the number of people who are eligible/ ineligible following the screening process. Specifically, recruitment will be considered feasible if more than 100 people respond and if at least half of the responders are eligible following screening. The feasibility of the measurement tools will first include whether participants have enough time to complete all measures. The feasibility of the questionnaires will also be assessed by any missing data. Additionally, the feasibility of the physiological measurements will be informed by the time taken to set up the equipment. The measures will, therefore, be considered feasible if they can all be completed in the allotted time (approximately 40 min).

## Acceptability

The acceptability of the study will be largely informed by the qualitative interview following the intervention.

Participants will be interviewed about their experience of the intervention via telephone approximately 1 week after they have completed the study. The interviews will be semistructured and follow a predefined schedule (see table 2). The interview will be centred around the acceptability of the intervention, but also aspects of the study. To this end, the interview will ask questions about motivations for taking part and expectations, how informed they felt before taking part, their experience of completing the questionnaires and having physiological measures taken, and their experience of the intervention itself. Some questions will also ask participants how they have felt since the intervention. Participants will then be given a chance to provide any other feedback or thoughts on taking part in the study. The audio from the interviews will be recorded and then transcribed.

**Table 2** Interview schedule for qualitative interviews

| Information and consent | 1. Were there any parts of the information sheet that were difficult to understand? |
| --- | --- |
| | 2. Did any part of your participation feel unexpected, based on what you were told in the information sheet? |
| Motivations for participating | 3. What motivated you to participate in the study? |
| | 4. What did you know about osteopathy before taking part? |
| Outcome measures— questionnaires | 5. Were there any questions or words on the questionnaires that were difficult to understand? |
| | 6. What was your experience like filling out the questionnaires? |
| Outcome measures— physiological | 7. What was your experience of having an ECG and blood pressure taken? |
| | 8. What was your experience of doing a heartbeat detection task? |
| Intervention | 9. Did you feel that the practitioner adequately explained the procedures to you? |
| | 10. What could have gone better during the session? |
| | 11. Did you take anything useful away from the session or learn anything new? |
| | 12. How likely are you to visit an osteopath again or seek similar treatments after this? |
| Closing points | 13. What else could you tell us about your experience of taking part in this study? |

Analysis of the data will be conducted using reflexive thematic analysis.[37] This involves initially familiarising oneself with the transcripts and then coding the data. Codes are then collated into themes. From here, themes are refined and categorised into main themes, midlevel themes and subthemes. Themes will then be discussed in terms of their strength. That is an indication will be provided of whether themes were common across many participants' accounts, or only mentioned by a few. It is hoped that by employing qualitative methods, a richer account of the acceptability of the study and intervention to participants can be obtained.

The practitioners will also be interviewed about their experience of delivering the intervention. This interview will also be thematically analysed, and the resulting themes explored.

Additionally, any adverse events occurring during the study will be logged using the adverse events report form (AERF; this can be found in online supplemental material 2).

## Psychological outcomes

These measures are intended to provide some initial data on the potential utility of the intervention for outcomes such as depression, anxiety and stress, psychological flexibility, and interoceptive awareness. They will be collected

during preintervention and postintervention and any changes will be analysed.

## Mental health
### Depression, Anxiety and Stress Scale

The DASS[38] is a self-report measure made up of 21 items with three subscales that measure depression, anxiety, and stress. The DASS will also be used as a screening tool to identify eligible participants in terms of the severity of mental health symptoms. Examples of items include 'I couldn't seem to experience any positive feeling at all' for the depression scale, 'I felt I was close to panic' for the anxiety scale and 'I found myself getting agitated' for the stress scale. These are then rated on a four-point Likert scale ranging from 0 (never) to 3 (almost always). Higher scores indicate higher levels of depression, anxiety and stress. The subscales have good internal reliability as measured by Cronbach's alpha coefficients ($\alpha$), which are 0.88 for depression, 0.82 for anxiety and 0.90 for stress, as well as 0.93 for the total score.[39]

### International PA and NA schedule-short-form

The International Positive and Negative Affect Schedule-Short-Form (PANAS-SF)[40] is a short-form version of the PANAS and uses 10 items to measure two subscales of PA and NA. Participants are asked to what extent they have felt certain states or emotions, such as 'inspired' for the PA scale and 'upset' for the NA scale. These are then rated on a five-point Likert scale ranging from 1 (very slightly or not at all) to 5 (extremely). Higher scores indicate higher levels of PA and NA. Both the PA and NA subscales have good internal reliability with both having a Cronbach's $\alpha$ of 0.84.[40]

## Psychological flexibility
### Acceptance and Action Questionnaire-II

The Acceptance and Action Questionnaire-II (AAQ-II)[41] is a self-report measure made up of seven items that measures psychological inflexibility or as it is also referred to, experiential avoidance. Items include a list of statements such as 'I'm afraid of my feelings' and 'worries get in the way of my success'. These items are then rated on a seven-point Likert scale from 1 (never true) to 7 (always true). Scores are then totalled with higher scores indicating greater levels of psychological inflexibility and experiential avoidance. The AAQ-II has good internal reliability with a Cronbach's $\alpha$ of 0.84.[41]

### Self as Context Scale

The Self as Context Scale (SACS)[42] uses 10 items to measure self-as-context, one of the acceptance components of psychological flexibility. Self-as-context can be described as a transcendent sense of self, where the individual is able to distance their 'noticing self' from internal thoughts and feelings. The SACS has two subscales: (1) centring, for example, 'when I am upset, I am able to find a place of calm within myself', and[1] transcending, for example, 'As I look back upon my life so far, I have a sense that part of me has been there for all of it'. Items

are then rated on a seven-point Likert scale from 1 (strongly disagree) to 7 (strongly agree). Higher scores on the subscales indicate higher levels of centring, and transcending and a higher total score indicates greater levels of self-as-context. The SACS has good internal reliability with Cronbach's $\alpha$ of 0.81 for centring, 0.78 for transcending and 0.81 for overall SACS score.[42]

## Interoceptive awareness
### Multidimensional Assessment of Interoceptive Awareness Version 2

The Multidimensional Assessment of Interoceptive Awareness Version 2 (MAIA-2)[43] is a 37-item self-report measure of interoceptive awareness. The MAIA-2 uses eight subscales which are (1): noticing, for example, 'when I am tense, I notice where the tension is located in my body',[1] not-distracting, for example, 'I distract myself from sensations of discomfort',[2] not-worrying, for example, 'when I feel physical pain, I become upset',[3] attention regulation, for example, 'I can pay attention to my breath without being distracted by things happening around me',[4] emotional awareness, for example, 'I notice how my body changes when I am angry',[5] self-regulation, for example, 'when I feel overwhelmed I can find a calm place inside',[6] body listening, for example, 'I listen for information from my body about my emotional state, and[7] trusting, for example, 'I trust my body sensations'. The items are rated on a six-point Likert scale ranging from 0 (never) to 5 (always). The scales have good internal reliability with the Cronbach's alpha coefficients for the scales being: 0.64 for noticing, 0.74 for not-distracting, 0.67 for not-worrying, 0.83 for attention regulation, 0.79 for emotional awareness, 0.79 for self-regulation, 0.80 for body listening and 0.83 for trust.[43]

## Physiological outcomes

These measures will provide initial data on how the intervention impacts psychophysiological factors. These measures will be collected during preintervention and postintervention and any changes analysed. The physiological measures are all being conducted in the same environment.

## Heart rate variability

HRV will be measured using a medical-grade Holter ECG monitor. Measurements will be taken at two time points, preintervention and postintervention. Participants will lie in a supine position while the ECG monitor records for at least 5 min. Participants will be asked in advance to refrain from consuming any caffeine, alcohol, or nicotine on the day of the study, to minimise interference with the ECG. A time-domain signal measure will be calculated using the root mean square of successive interval differences (RMSSD). Frequency-domain measurements will also be calculated by using low-frequency (LF) power, high-frequency (HF) power and LF to HF ratio. This measure will be analysed in conjunction with the recent literature that suggest it is a measure of primarily the parasympathetic system.[44]

### Interoceptive accuracy

Participants will perform a heartbeat detection task as a measure of IAc. This is conducted in the form of the heartbeat perception task which is performed according to the Mental Tracking Method[45] using intervals of 30, 35, 40 and 45s that are separated by 30s resting periods. During each trial R–R intervals are recorded, and participants are asked to silently count their heartbeats without the use of an exteroceptive aid (such as taking one's pulse). At the end of each period, participants verbally report the number of counted heartbeats. The participants will not be informed about the length of the counting phases nor the quality of their performance. Interoceptive sensibility will also be measured through participants' subjective assessments about how accurately they perceived heartbeats.[46] These measures will be completed preintervention and postintervention.

### Blood pressure

BP will be measured at preintervention and postintervention. This will be carried out in line with the National Institute for Health and Care Excellence (NICE) recommendations. That is, BP will be collected in a room that is quiet, relaxed and temperate, while the participant will be quiet and seated, and their arm outstretched and supported, using an appropriate cuff size for the person's arm.[47] This outstretching of the arm will allow the practitioner to assess any undiagnosed HBP. If the participant has HBP it can make some of the osteopathic techniques less safe,[35 36] so it is important to establish this. HBP will be determined according to the NICE recommendation of BP results that are 140/90 mm Hg and over.[47] In addition to participant safety, measuring BP will provide data on any impact the intervention might have on this physiological indicator.

### Additional outcomes

Demographic information will also be collected from participants relating to their gender, age and ethnic background. Although participants will have been screened for chronic pain, they will be asked whether they are currently or have recently been experiencing any neck pain or headaches. This is to inform the clinician about any problematic body areas, which may, therefore, be avoided in the intervention. The participants should be presenting as pain-free due to the initial screening process, but this is still a necessary safety measure. Participants will be excluded from the analysis if they present with neck pain or headaches.

Participants will also be asked whether they are currently receiving any mental health treatment. They will be asked whether they are currently taking any antidepressants or other related prescribed medication for mental health issues. Participants will also be asked whether they have recently attended or are currently attending any form of talking therapy or other psychotherapy. Participants' prescription medication or psychotherapy status will not exclude them from the study. However, this will again be entered as a covariate if several participants report that they are receiving these psychological treatments.

Lastly will be the noting of any adverse events that occur during the intervention or study period. These will be filled out by the practitioner using the AERF and collected by the researcher if occurring during the intervention. If participants contact the researcher after the intervention regarding an adverse event, then this will be logged by the researcher.

### Participant timeline

See table 3 for the participant timeline.

### Sample size

The study will aim to recruit 32 participants. This number of participants is generally deemed sufficient for feasibility studies[48] and would represent approximately 10% of the sample size required in a full trial.[49] This sample size also falls within what is practical given the available resources.

### Recruitment

Recruitment at the university is being conducted by advertising in communal spaces with posters. Additionally, social media will be used for recruitment by reaching out to mental health support groups and sharing an advertisement for the study on various social networks (note: this recruitment work has begun). Participants will contact the research team if they are interested in taking part. They will then be given an information sheet to read and a consent form to sign. Following this, they will complete the DASS to complete to determine their eligibility regarding mental health symptoms. The cutoff scores for mild to moderate will be defined as follows: depression=10–20, anxiety=8–14 and stress=15–25.[50] If eligible they will be invited to take part in the intervention. If they display severe mental health symptoms, they will not be invited to take part further and signposted to the relevant mental health services and charities.

## METHODS: ASSIGNMENT OF INTERVENTIONS
### Allocation
#### Sequence generation

Thirty-two participants will be randomly assigned to one of the four conditions using a computerised random number generator. Permuted block randomisation will be used to ensure that equal numbers of participants are in each condition. The block sizes will not be disclosed to help ensure concealment and prevent any potential prediction of group allocation. This will be conducted by the principal investigator (PI) of the study DJE, while the outcome assessor JH-B is blinded to this randomisation process.

#### Concealment mechanism

Allocation concealment will be ensured using sequentially numbered, sealed opaque envelopes which contain the group assignment. The PI will carry out the allocation

**Table 3** Participant timeline

| Activity/ assessment | Approx. time to complete | T$_{-1}$ Prestudy screening/ consent | T$_0$ Prestudy randomisation | T$_1$ Preintervention | T$_2$ Intervention | T$_3$ Postintervention tests | F$_1$ Follow-up 1 week |
|---|---|---|---|---|---|---|---|
| Informed consent | 5 min | X | | | | | |
| Screening with DASS | 5 min | X | | | | | |
| Randomisation | 15 min | | X | | | | |
| Baseline assessment— questionnaires | 15 min | | | X | | | |
| Baseline assessment— physiological | 15 min | | | X | | | |
| Intervention | 30 min | | | | X | | |
| Postintervention questionnaires | 15 min | | | | | X | |
| Postintervention physiological | 15 min | | | | | X | |
| Telephone interview | 30 min | | | | | | X |

DASS, Depression Anxiety Stress Scale; F, follow-up; T, timepoint.

concealment, and ensure that the outcome assessor is blinded to the intervention allocation.

## Implementation

All participants who provide informed consent and who meet the eligibility criteria will be randomised into a study condition (as described in section Sequence generation). The randomiser DJE will not be directly involved in the recruitment or data collection, and instead, the outcome assessor will conduct the recruitment. The list of random numbers that correspond to group allocation will not be revealed to the researcher (JH-B) involved in data collection or recruitment. The sealed envelopes will contain a randomisation number and corresponding intervention identity code for the allocation of participants into the intervention groups. The osteopathic practitioner will then be able to open the envelope and determine which intervention is to be delivered on the day the study is conducted.

## Blinding (masking)

The outcome assessor will be blind to the participant's group allocation. After preintervention psychometric and psychophysiological measures (see sections Psychological outcomes and Physiological outcomes, respectively) have been completed, the outcome assessor will leave the room (to ensure blinding) and the intervention will begin, conducted by the osteopath. Participants will not be blinded to study intervention, as the osteopathic practitioner will need to explain study and intervention procedures, in line with the osteopathic practice standards and

ethical consent.[13] The practitioner will not be blinded to the intervention type (as they need to know what intervention to deliver) but will be blinded to the study outcomes. The outcome assessor will also be conducting the data analysis, and the random numbers corresponding to each group will only be revealed when this analysis has been completed. To ensure participants do not disclose the condition they were allocated to, they will be asked not to communicate directly to the outcome assessor about the intervention they received. The study will therefore be single-blinded, where the outcome assessor is blind to intervention allocation, and the osteopathic practitioner will not be blind (hence single-blind).

## Emergency unblinding

As the practitioner is not blinded, no emergency unblinding procedures are deemed necessary.

## METHODS: DATA MANAGEMENT AND ANALYSIS
### Data management

All data will be entered electronically at the university where the data are being collected and kept in a password-protected folder, which only the outcome assessor will have access to for the duration of the study. The electronic data will be kept confidential, and participants' names will not be linked to their datasets. For the longer term, electronic datasets will be kept indefinitely in the interest of transparency to fulfil any requests for the original data and maintained on the OSF.

## STATISTICAL METHODS
### Outcomes

Statistical analysis will be conducted using IBM SPSS (V.27). Means and SD will be reported for demographic data that includes gender, age and ethnicity. For the main analysis, data will first be examined for normality using the Shapiro-Wilk test. If data are skewed, logarithmic transformation will be used, otherwise, analysis will continue without any transformation. HRV data will be preprocessed, and inspected for any potential artefacts, and these will be removed if identified. RMSSD will be calculated on the preprocessed artefact removed data using Kubios V.3.5 (https://www.kubios.com/) via Matlab V.R2021a. IAc will then be calculated using the formula: $IAc = 1/4 \Sigma [1-(|recorded\ heartbeats-counted\ heartbeats|/recorded\ heartbeats)]$. The psychometrics will be totalled according to the relevant questionnaire instructions and subscales.

The main analysis will comprise of seven separate mixed design two (preintervention and postintervention) by four (AVHT, STM, CST, combination (please see: Interventions section for full details of these interventions.)) analysis of covariance (ANCOVA) models. This will comprise of five separate ANCOVAs for the five psychometrics (DASS, PANAS-SF, AAQ, SACS, MAIA) and another two ANCOVAs for the physiological measures of IAc and HRV (as measured by RMSSD and LF/HF ratio). Covariates will consist of (1) whether participants are currently receiving psychotherapy (yes or no) and (2) whether participants are receiving pharmacological treatment (yes or no). Significant models will be examined further using post hoc Bonferroni tests.

### Additional analyses

Exploratory correlational analyses will also be conducted to examine relationships between changes from preintervention to postintervention on the various measures (eg, the relationship between change from pre–post HRV RMSSD and pre–post DASS scores).

### Analysis of population and missing data

The study will operate on an intention-to-treat basis. All participants randomised and with preintervention data will be included in the final analysis. Any participants with missing data will be included in the analysis using the multiple imputation feature of SPSS.

## METHODS: MONITORING
### Data monitoring

As this study is taking place over a short duration as a feasibility study and not a full RCT, no formal committee for data monitoring is required.

### Harms

The osteopathic practitioner will inform the participants about the general potential common adverse effects of osteopathy namely some stiffness and soreness in the days following the intervention, and rare adverse events including tissue damage,[51] in line with informed consent processes. The osteopathic practitioner will record any adverse effects on the day the intervention is received (that occur during or immediately after the intervention) in the AERF (see online supplemental material 2). Participants will also be advised to contact the PI DJE by telephone if they have any concerns or adverse events following the intervention in subsequent days after the intervention was received. If such events are reported, these will again be reported by DJE in the AERF. Any adverse events or harms that are ranked highly on severity will be reported to the ethical committee. This includes any adverse events that, for example, require hospitalisation.

### Auditing

As the study is taking place over a short duration and only at one site, no formal auditing processes are deemed necessary, though PI will have regular team meetings to ensure the study is following the research protocol at all times.

## METHODS: PATIENT AND PUBLIC INVOLVEMENT STATEMENT

Key stakeholders were consulted and involved at a very early stage of the research process. The Patient Experience and Evaluation in Research (https://www.swansea.ac.uk/humanandhealthsciences/research-at-the-college-of-human-and-health/patientexperienceandevaluationinresearchpeergroup/) group in the College of Human and Health Sciences at Swansea University were consulted. This group represented members of the public, students, and staff members, several of whom reported that they had experienced depression, anxiety or stress at some point in their lives and emphasised the need for innovative approaches to the delivery of mental health support. The feasibility design was explained to them, and they gave positive feedback about the nature of the preliminary research plan.

## ETHICS AND DISSEMINATION
### Research ethics approval

The protocol for this feasibility study has received ethical approval from the Department of Psychology Ethics Committee at Swansea University, ethical review reference number: 2022-5603-4810.

### Protocol amendments

Any deviations from the protocol that could impact the conduct or bias of the study will be clearly outlined and justified in the final written report. Version control of the protocol using identifiers and dates, along with a list of amendments will be clearly listed. This will enable tracking of the history of amendments and identification of the most recent protocol version.

## Consent

Participants will scan a QR code on recruitment posters or click a link via email/social media adverts that will take them to the study's information sheet. The information sheet emphasises that participation is voluntary and that they can withdraw from the study at any stage, without needing to provide a reason. If they have any questions or concerns at this stage, they are encouraged on the information sheet to contact the research team. If they are willing to proceed, they will complete an online consent form (see online supplemental material 3).

## Ancillary research

The data collected in this study will not be used for any other ancillary research.

## Confidentiality

Participants will be assigned a coded ID number to maintain confidentiality. Any records of personal identifiers such as informed consent forms will be stored separately from data with ID numbers. To limit data access to the minimum number of individuals, only the researcher JH-B will have access to the data for analysis.

## Declaration of interests

The individual authors have no direct conflicts of interest to declare.

## Access to data

Only the researcher JH-B will have access to the dataset during the study period. On completion, the collected data will be deidentified and made available on the OSF. Similarly, the SPSS statistical syntax code used will be made available on OSF.

## Ancillary and post-trial care

Participants will be fully debriefed once they have completed the study. The contact details of the research team will be provided should participants have any concerns. As the participants will be presenting with mild to moderate mental health symptoms, the debrief form will encourage participants to seek support services such as mental health charities or their general practitioner if their psychological condition deteriorates at any time.

## DISSEMINATION POLICY
### Trial results

Following the completion of the study, it is anticipated to take around 2–3 months to compile the final results ready for publication in an appropriate peer-reviewed journal. The study's results may also be used as part of presentations at any relevant conferences.

## Authorship

The authors of this protocol will also be the authors of the final report. All authors have made substantive contributions to the design of the study. Additionally, all authors will have made substantive contributions to the interpretation of the data collected and the writing of the final report.

## Reproducible research

This protocol will be available to researchers via open-access publication. The dataset collected will be deidentified and made available on OSF. Similarly, the statistical syntax code used will be made available on OSF. These will be made available no later than 1 year on completion of data collection.

**Contributors** JH-B wrote the first draft of this paper, then assisted with subsequently revising additional drafts. JH-B also made substantive contributions to the concept, design and writing of this study. JD-R assisted with revising additional drafts of this paper, and also made substantive contributions to the concept, design and writing of this study. DJE was the principal investigator on the grant (Osteopathic Foundation) that funded this work. DJE, therefore, made substantial contributions to the design, concept and writing of this study.

**Funding** This research has been funded by The Osteopathic Foundation, grant award number: URNLG010.

**Disclaimer** The funders have no direct role in conducting this study. The funding is primarily being used to pay for the role of the research assistant on this project, held by JH-B.

**Competing interests** None declared.

**Patient and public involvement** Patients and/or the public were involved in the design, or conduct, or reporting, or dissemination plans of this research. Refer to the Methods section for further details.

**Patient consent for publication** Not applicable.

**Provenance and peer review** Not commissioned; externally peer reviewed.

**ORCID iDs**
Jerry Draper-Rodi http://orcid.org/0000-0002-1900-6141
Darren J Edwards http://orcid.org/0000-0002-2143-1198

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
