## [Reviewer comments · BMJ Open]

ARTICLE DETAILS

TITLE (PROVISIONAL)	Applying an osteopathic intervention to improve mild to moderate mental health symptoms: a mixed-methods feasibility study protocol
AUTHORS	Hope-Bell, Josh; Draper-Rodi, Jerry; Edwards, Darren

VERSION 1 – REVIEW

REVIEWER	Bohlen, Lucas Osteopathie Schule Deutschland GmbH, Osteopathic Research Institute
REVIEW RETURNED	15-Feb-2023

GENERAL COMMENTS	I congratulate the authors on the manuscript. This study protocol addresses a relevant gap in the literature and applies sound methods (adhering to the SPIRIT guideline). Still, there are minor concerns that need to be addressed. In general, there are some issues with the references, page numbers, typos, abbreviations, and missing material. First, the reference style is not coherent throughout the manuscript as some citations are presented by author names and others by numbers. Second, the page numbers are not correctly formatted (after page 9). Although not consecutively formatted in the manuscript, this peer-review report references the original page numbers (continued after page 9). Third, there appear to be typos on page 6 (Lines 19-24), page 10 (Lines 6-10), page 19 (Lines 48-52), page 23 (Lines: 37-42), and page 25 (Lines 35-38). Fourth, a few abbreviations are not introduced at first usage (Page 11, Lines 47-50; Page 20, Lines 15-20). Fifth, supplementary material 2 seems to be missing (Page 25, Lines 12-15). The abstract would benefit from specifying the number of treatment sessions, defining the methods used for the qualitative and quantitative analysis, and clearly stating the psychological outcomes (e.g., psychological flexibility). In the introduction, the rationale for linking mental health and physiological outcomes should be outlined (the authors provided such information in previous work they cited: https://osf.io/jrtpx/?view_only=63ffa916b76c4b95b4233d3cd812f12d). Furthermore, the last paragraph of the background and rationale section (6.1) may be extended to provide further information (citing additional relevant studies) on the general effect of osteopathic treatment on the physiological and psychological outcomes of interest (i.e., mental health outcomes as well as autonomic and interoceptive markers). Subsequently, the specific effect of osteopathic techniques (used in the study) on these measures was outlined (6.2), which
---

	should clearly state if no evidence is available (e.g., for massage and interoception). The objectives should be clearly described and linked to the outcomes and measures. Further, the objectives do not mention blood pressure, which should be added to the hypothesis for the secondary objective (Page 7, Lines 31-36). The trial design should be identified as a feasibility study. Furthermore, please explain why the allocation ratio is 1:1 if there are four conditions and no control groups (Page 7, Lines 47-50). The eligibility criteria define that subjects must report mild and moderate (but not severe) mental health symptoms (Page 8, Lines 22-24). However, it is only later mentioned that the DASS-21 will be used to screen subjects for eligibility (Page 13, Lines 17-20). Furthermore, no diagnostic cut-off was defined for the DASS-21 to differentiate between 'no', 'mild', 'moderate', and 'severe' mental health symptoms (Page 19, Lines 47-57). The intervention section would benefit from further information about the therapist applying the techniques (e.g., age and gender of the osteopath as well as years of practice experience) (Page 8, Lines 42-52). Further, it may be scrutinized if the main target of craniosacral techniques are muscles around the head and spine (although it may be true, conceptually, the cranial bones would arguably be the main target) (Page 10, Lines 56-59). Please provide a justification and reference for not applying high-pressure techniques in patients with high blood pressure, neck pain, or headaches (Page 11, Lines 47-59; Page 16, Lines 17-20). Lastly, it may be advantageous to exclude subjects if they change their concomitant care (e.g., terminate the use of antidepressants during the trial). The reporting structure for the outcomes (and objectives) could be improved. Please define each outcome clearly, stating the name (e.g., psychological flexibility), measure (e.g., self as context scale), analysis (e.g., pre- to post-intervention), type (e.g., psychological outcome), and priority (e.g., secondary outcome). Specifically, the feasibility and acceptability outcomes must be described in more detail (Page 12, Lines 24-59), and it should be defined what constitutes feasibility compared to infeasibility. Furthermore, the introductory paragraph for the psychological outcomes (Page 13, Lines 5-10), may also be added to the physiological outcomes (Page 15, Line 20). More details about the HRV collection methods are needed; for example, will HRV be recorded in a supine position during a five-minute period? (Page 15, Lines 24-34). The authors may also consider assessing body mass index as a covariate (Page 16, Line 50). If the sample should be pain-free (Page 3, Lines 8-10), subjects reporting pain after the initial screening process (e.g., neck pain and headaches) should arguably be excluded (not used as a covariate) (Page 16, Lines 33-50). Please specify how the subjects will contact the investigators to report adverse events (e.g., by phone) (Page 17, Lines 15-17; Page 23, Lines 43-52). It should also be specified if the interview schedule applies to both the subjects and the therapist (Page 18, Lines 8-10). The examples of questions for each subscale should be reported consistently in either the text or brackets (Page 14, Lines 56-59). The abstract states that recruitment rates are evaluated as additional outcomes (Page 2, Lines 33-36), however, they are not thematized in the main text and should arguably be part of the primary outcomes. Lastly, it is not clear, why
--	--

	the qualitative outcomes (12.6) are not reported in the acceptability section (12.2), although acceptability will be rated largely based on the interviews (Page 12, Lines 45-47). Part of the recruitment process was described in the study setting section (Page 8, Lines 4-13), but may be more appropriately placed in the recruitment section (Page 19, Lines 43-57). Harms that are common to osteopathic treatment may require a reference (Page 23, Lines 31-38). Further, it is unclear how the severity of adverse events will be defined (Page 23, Lines 50-52). The consent section refers to supplementary material 2 (Page 25, Lines 12-15), which seems to be missing from the manuscript and submitted materials. Information on competing interests and funding sources are outlined multiple times (Page 3, Lines 47-50; Page 4, Lines 12-15; Page 25, Lines 33-38; Page 27, Lines 3-8; Page 27, Lines 10-15), which appears redundant.
--	---

REVIEWER	Keller, Micha RWTH Aachen University
REVIEW RETURNED	08-Mar-2023

GENERAL COMMENTS	The manuscript “Applying an osteopathic intervention to improve mild to moderate mental health symptoms: a mixed-methods feasibility study protocol” submitted to BMJ open by Hope-Bell and colleagues presents a study protocol using three standard types of osteopathic interventions (+combine condition) to investigate their effect on psychophysiological and mental health variables. The study is based on evidence that osteopathic interventions may also affect mental health symptomatology, e.g., by improving psychophysiological functioning indicated by HRV. Overall, the presented protocol (recruitment started in December 2022) makes a sound impression and I’m interested in the final study results. Please find some detailed comments below. In section 6.2 the references are used in a different style and some of them (e.g., Castro-Sanchez, 2014, Sherman et al., 2010) cannot be found in the references list. This makes it difficult to see which studies were actually referenced. Most of the text is well written, however, there are some minor typos, missing commas and it makes sense to check carefully again (e.g., p. 11: The practitioner will first observe the participant while standing, then will observe active range of movements in with the participant in standing and/or sitting positions.). Furthermore, the description of psychophysiological literature could be a bit more differentiated as results concerning, e.g., HRV are not always unidirectional and interpretation often difficult. As feasibility of the interventions is the main concern of the study, this can be excused. In section 7.0 Objective: “symptoms of mental health” should be rather “mental health symptomatology” or “symptoms of mental illness”? Why not use 32 participants, a number which can be divided by 4 groups? The authors decided not to use a pure (time) control group, but to rather compare several potentially effective osteopathic interventions as the focus is on the feasibility of applying these to patients with mild and moderate mental health issues. However, are group differences expected or are all interventions thought to have the
--

	same effect? Concerning section 12.4.1 HRV: It is not defined how long the measurement will be. Is there a baseline? Are participants seated? Are these measurements collected in the same environment as BP measurements? Furthermore, are participants asked to refrain from caffeine, alcohol, ... as these are factors that could potentially influence psychophysiological measurements? One of the measures used by the authors is the LF/HF quotient. Considering difficulty of interpretation (LF is thought to be influenced by sympathetic as well as parasympathetic ANS) of this measure (see e.g., Reyes del Paso, G. A., Langewitz, W., Mulder, L. J., Van Roon, A., & Duschek, S. (2013). The utility of low frequency heart rate variability as an index of sympathetic cardiac tone: a review with emphasis on a reanalysis of previous studies. Psychophysiology, 50(5), 477-487.) it may be a good idea to rethink HRV analyses.
--	---

REVIEWER	Esteves, Jorge Malta International College Osteopathic Medicine
REVIEW RETURNED	03-May-2023

GENERAL COMMENTS	Background and Rationale Overall, the background and rationale section provides a clear introduction to the topic of mental health problems and the potential use of osteopathic interventions as a complementary approach. The section effectively highlights the increasing burden of mental health problems in the UK and the limitations of traditional forms of care. The rationale for considering innovative approaches, such as osteopathy, is well justified. However, there are a few areas that could be improved or clarified: In line 27, the statement "All of the techniques used have an effect on the interplay between the nervous and musculoskeletal systems" could benefit from further elaboration or citation to support this claim. Providing specific examples or evidence for how osteopathic techniques influence the nervous and musculoskeletal systems would strengthen the argument. In line 58, the reference to a pre-registered systematic review on OSF is provided, but it would be beneficial to include a brief overview of the protocol and the expected outcomes or objectives of the review. By addressing these points, the background and rationale section would be further strengthened in terms of providing clear justification for the study and enhancing the credibility of the presented information. Objectives The objective is clear and concise, stating the purpose of the study to assess the feasibility and acceptability of four osteopathic interventions for adults with mild to moderate symptoms of mental health. The secondary aim is well-defined, indicating that the study also aims to assess the impact of the four interventions on physiological factors and their effectiveness in improving psychological outcomes. The hypothesis is straightforward and aligns with the objective of assessing feasibility and acceptability.
--

	The hypothesis is clear, stating that the interventions will lead to psychophysiological relaxation by increasing heart rate variability (HRV) and improving interoceptive accuracy. The final hypothesis is well-stated, indicating that the four interventions will result in improvements in measures of mental health. Overall, the objectives and hypotheses are clearly articulated, providing a comprehensive understanding of the study's goals and expectations. Trial design The description of the trial design is clear, stating that an explanatory sequential mixed-methods approach will be used, with the quantitative aspect preceding the qualitative aspect. The description of the quantitative aspect of the trial design is concise and effective, indicating that a parallel, randomized design will be used with a 1:1 allocation to each of the four conditions. The qualitative aspect of the trial design is well-explained, mentioning that interviews will be conducted with both the participants of the intervention and the practitioners delivering them. The trial design section provides a clear overview of the approach, highlighting the use of both quantitative and qualitative methods in a sequential manner. The allocation method for the quantitative aspect is also specified. Methods: Participants, interventions, and outcomes. Study setting The study setting is clearly stated as Swansea University in South Wales, UK. The recruitment sources are also mentioned, including the student population and the general public, with specific methods such as advertising in communal spaces and utilizing social media. The information regarding the location and start date of the study is provided concisely. Eligibility Criteria The eligibility criteria are clearly defined, specifying the age range, target symptoms, and language proficiency required for participation. The rationale for excluding participants experiencing pain is provided, explaining the potential confounding effects on mental health outcomes. Interventions The description of the interventions is clear, stating the four types of osteopathic techniques and providing an estimated duration for each session. Referring to Table 1 for a summary adds clarity. Modifications The mention of potential modifications to ensure participants' safety and avoid discomfort is appropriate and responsible. Adherence The explanation regarding adherence is appropriate for a single-session intervention, and the plan to record any instances of participants ending the session early adds transparency. Concomitant care
--	--

	The plan to collect information about participants' concomitant care and account for it in the statistical analysis is appropriate and ensures that these factors are considered during the study. Overall, the Methods section provides detailed information about the study setting, eligibility criteria, interventions, modifications, adherence, and concomitant care. The descriptions are clear and provide sufficient information for understanding the study design. Outcomes The section provides an overview of the outcomes that will be assessed in the study. It includes subsections on feasibility, acceptability, psychological outcomes, physiological outcomes, additional outcomes, and qualitative outcomes. The subsections provide clear descriptions of the measures and procedures that will be used to assess each outcome. The information is well-organized and easy to follow. Overall, this section provides a comprehensive overview of the outcomes that will be evaluated in the study. Feasibility: The subsection describes the feasibility of the recruitment process and the measurement tools. It mentions that feasibility will be determined by the number of people who respond to advertisements and the number of people who are eligible/ineligible following the screening process. It also mentions that the feasibility of the measurement tools will be assessed based on participants' ability to complete the measures and the presence of missing data. The subsection provides a clear explanation of how feasibility will be evaluated. No specific issues were identified in this subsection. Acceptability The subsection provides a clear description of the methods for assessing acceptability. No specific issues were identified in this subsection. Psychological outcomes The subsection provides clear and concise information about the measures. No specific issues were identified in this subsection. Physiological outcomes The subsection provides clear information about the physiological measures. No specific issues were identified in this subsection. Additional outcomes The subsection provides a clear explanation of why these outcomes are being assessed and how they will be collected. No specific issues were identified in this subsection. Qualitative outcomes The subsection provides clear information about the qualitative data collection and analysis process. No specific issues were identified in this subsection. Overall, the sections are well-written and provide a comprehensive overview of the outcomes that will be assessed in the study. The descriptions are clear and concise, making it easy to understand the measures and procedures that will be used. The subsections are logically organized and flow well. No major issues were identified in the sections reviewed.
--	--

	Sample Size This section provides a clear and concise explanation of the sample size, but it would be helpful to mention what the previous research that informs the decision was and cite the sources. Recruitment This section describes the recruitment process for participants, which involves contacting the research team, reading an information sheet, and completing the Depression, Anxiety, Stress Scale (DASS) to determine eligibility. The authors also mention that participants with severe mental health symptoms will be signposted to relevant mental health services and charities. The section is well-written and provides a clear and concise explanation of the recruitment process. Allocation This section is clearly written and provides a detailed explanation of the randomization process, but it would be helpful to mention the sample size again and explain why four conditions were chosen. Blinding (masking) This section explains how blinding will be ensured for the outcome assessor who will be blind to the participants' group allocation. The authors also state that the osteopathic practitioner will not be blinded to the intervention type but will be blinded to the study outcomes. The section is well-written and provides a clear explanation of the blinding process. Data management This section is clear and concise and provides a detailed explanation of data management. Statistical Methods This section briefly describes the statistical methods that will be used to analyze the data. Overall, the sections of the paper reviewed are well-written, clearly explained, and provide a detailed description of the study design and methods. However, there are some areas where more information could be provided, such as the statistical methods used and why four conditions were chosen for the study. Additionally, citing the sources for the previous research that informed the sample size decision would strengthen the study's credibility.
--	---

VERSION 1 – AUTHOR RESPONSE

Reviewer: 1

Reviewer 1> I congratulate the authors on the manuscript.

Authors> We thank the reviewer for these kind words.

Reviewer 1>This study protocol addresses a relevant gap in the literature and applies sound methods (adhering to the SPIRIT guideline). Still, there are minor concerns that need to be addressed. In general, there are some issues with the references, page numbers, typos, abbreviations, and missing material. First, the reference style is not coherent throughout the manuscript as some citations are

presented by author names and others by numbers. Second, the page numbers are not correctly formatted (after page 9). Although not consecutively formatted in the manuscript, this peer-review report references the original page numbers (continued after page 9). Third, there appear to be typos on page 6 (Lines 19-24), page 10 (Lines 6-10), page 19 (Lines 48-52), page 23 (Lines: 37-42), and page 25 (Lines 35-38). Fourth, a few abbreviations are not introduced at first usage (Page 11, Lines 47-50; Page 20, Lines 15-20). Fifth, supplementary material 2 seems to be missing (Page 25, Lines 12-15).

Authors> Thank you for identifying these formatting errors. The referencing has now been updated so it is consistently in Vancouver style. The page numbering has been updated so that it is now consecutive and correct. The typos have now been corrected. The abbreviations have now been introduced correctly. Lastly, supplementary material 2 is now included in the revised version.

Reviewer 1> The abstract would benefit from specifying the number of treatment sessions, defining the methods used for the qualitative and quantitative analysis, and clearly stating the psychological outcomes (e.g., psychological flexibility).

Authors> Many thanks for this helpful suggestion, we have now included this information in the abstract as requested:

- “receive a single intervention treatment session”
- “five standardised questionnaires, which include: (1) the Depression, Anxiety and Stress Scale; (2) the International Positive and Negative Affect Schedule- Short-Form; (3) Acceptance and Action Questionnaire-II ; (4) the Self as Context Scale; (5) and the Multidimensional Assessment of Interoceptive Awareness Version 2”
- “Mixed design two (pre- and post-intervention) by the four interventions analysis of covariance (ANCOVA) models will be used to analyse the quantitative data for each quantitative measure. Participants will also be interviewed about their experiences of the study and interventions and a thematic analysis will be used to analyse this qualitative data.”

Reviewer 1> In the introduction, the rationale for linking mental health and physiological outcomes should be outlined (the authors provided such information in previous work they cited: https://eur03.safelinks.protection.outlook.com/?url=https%3A%2F%2Fosf.io%2Fjrtpx%2F%3Fview_only%3D63ffa916b76c4b95b4233d3cd812f12d&data=05%7C01%7CD.J.Edwards%40Swansea.ac.uk%7Ccb4501456dbe4355553808db4d5933a2%7Cbbcab52e9fbc43d6a2f39f66c43df268%7C0%7C0%7C638188817738034353%7CUnknown%7CTWFpbGZsb3d8eyJWljoimc4wLjAwMDAiLCJQIjoiV2luMzliLCJBTil6lk1haWwiLCJXVCi6Mn0%3D%7C3000%7C%7C%7C&sdata=v5RW8h%2Fhk5d9jrgFIFBJ1dSkqDNFbhwG4h9%2F8mr3OHU%3D&reserved=0). Furthermore, the last paragraph of the background and rationale section (6.1) may be extended to provide further information (citing additional relevant studies) on the general effect of osteopathic treatment on the physiological and psychological outcomes of interest (i.e., mental health outcomes as well as autonomic and interoceptive markers). Subsequently, the specific effect of osteopathic techniques (used in the study) on these measures was outlined (6.2), which should clearly state if no evidence is available (e.g., for massage and interoception).

Authors> Many thanks for this very helpful suggestion. We have now included in the rationale some of the specific citations that expand on the general effects of osteopathic techniques that we use in this protocol for the measure we have outlined in 6.2, as you request.

pp. 5-6 “This also includes a rationale for linking mental health with physiological mechanisms, as previous studies have examined how osteopathy may influence psychophysiological factors. A number of these have examined the influence of osteopathic manipulative therapy (OMT) on heart rate variability (HRV), which is considered a potentially important indicator of physical and psychological wellbeing (15, 16). Cerritelli et al. (17) found that two sessions of OMT significantly increased HRV in healthy adults, relative to a sham control group. Similarly, Arienti et al. (18) found that applying a single session fourth ventricle compression (CV4) technique significantly increased HRV, compared to a placebo intervention. So, there seems to be some clear evidence that influencing physiological mechanisms through osteopathy are highly relevant for mental health improvement.”

Reviewer 1> The objectives should be clearly described and linked to the outcomes and measures. Further, the objectives do not mention blood pressure, which should be added to the hypothesis for the secondary objective (Page 7, Lines 31-36).

Authors> Thank you for this feedback, the objectives have now been more clearly linked to specific measures and blood pressure is included in the hypothesis section.

Reviewer 1> The trial design should be identified as a feasibility study. Furthermore, please explain why the allocation ratio is 1:1 if there are four conditions and no control groups (Page 7, Lines 47-50).

Authors> Thank you for this comment, the section now clearly mentions the feasibility study design. The sample size has been updated to 32 participants so that a 1:1 allocation is achieved.

Reviewer 1> The eligibility criteria define that subjects must report mild and moderate (but not severe) mental health symptoms (Page 8, Lines 22-24). However, it is only later mentioned that the DASS-21 will be used to screen subjects for eligibility (Page 13, Lines 17-20). Furthermore, no diagnostic cut-off was defined for the DASS-21 to differentiate between 'no', 'mild', 'moderate', and 'severe' mental health symptoms (Page 19, Lines 47-57).

Authors> Thanks for raising this. The DASS is now clearly mentioned as the screening tool in the eligibility criteria section. Additionally, the cutoff scores have now been added to section 15, page 21. p. 21: "The cut-off scores for mild to moderate will be defined as follows: depression = 10-20, anxiety = 8-14, and stress = 15-25 (46)."

Reviewer 1> The intervention section would benefit from further information about the therapist applying the techniques (e.g., age and gender of the osteopath as well as years of practice experience) (Page 8, Lines 42-52). Further, it may be scrutinized if the main target of craniosacral techniques are muscles around the head and spine (although it may be true, conceptually, the cranial bones would arguably be the main target) (Page 10, Lines 56-59). Please provide a justification and reference for not applying high-pressure techniques in patients with high blood pressure, neck pain, or headaches (Page 11, Lines 47-59; Page 16, Lines 17-20). Lastly, it may be advantageous to exclude subjects if they change their concomitant care (e.g., terminate the use of antidepressants during the trial).

Authors> Thanks for these helpful suggestions, more information about the osteopaths has now been added:

p. 9 "The interventions are being delivered by two male osteopaths, one with 17 years of practice experience and one with 3 years of practice experience.

The target of CST is difficult as foundational work in craniosacral approaches were focused on bones, their supposed movements and cerebrospinal fluid dynamics. The clear mechanisms involved in craniosacral techniques are unclear, but the theory used in this trial was based on Gabutti & Draper-Rodi's proposal that the techniques may be more related to soft tissue changes than to deeper structures (ref 35).

Additionally, justification for not applying HVT in the presence of HBP and supporting references have now been added:

36. Rushton A, Carlesso LC, Flynn T, Hing WA, Rubinstein SM, Vogel S, et al. International Framework for Examination of the Cervical Region for Potential of Vascular Pathologies of the Neck Prior to Musculoskeletal Intervention: International IFOMPT Cervical Framework. *Journal of Orthopaedic & Sports Physical Therapy*. 2022;53(1):7-22.

37. Vaughan B, Moran R, Tehan P, Fryer G, Holmes M, Vogel S, et al. Manual therapy and cervical artery dysfunction: Identification of potential risk factors in clinical encounters. *International Journal of Osteopathic Medicine*. 2016;21:40-50.

The reviewer makes a valid point regarding concomitant care, however as the trial period is so brief this is unlikely to be an issue.

Reviewer 1> The reporting structure for the outcomes (and objectives) could be improved. Please define each outcome clearly, stating the name (e.g., psychological flexibility), measure (e.g., self as context scale), analysis (e.g., pre- to post-intervention), type (e.g., psychological outcome), and priority (e.g., secondary outcome). Specifically, the feasibility and acceptability outcomes must be described in more detail (Page 12, Lines 24-59), and it should be defined what constitutes feasibility compared to infeasibility. Furthermore, the introductory paragraph for the psychological outcomes (Page 13, Lines 5-10), may also be added to the physiological outcomes (Page 15, Line 20). More details about the HRV collection methods are needed; for example, will HRV be recorded in a supine

position during a five-minute period? (Page 15, Lines 24-34). The authors may also consider assessing body mass index as a covariate (Page 16, Line 50). If the sample should be pain-free (Page 3, Lines 8-10), subjects reporting pain after the initial screening process (e.g., neck pain and headaches) should arguably be excluded (not used as a covariate) (Page 16, Lines 33-50). Please specify how the subjects will contact the investigators to report adverse events (e.g., by phone) (Page 17, Lines 15-17; Page 23, Lines 43-52). It should also be specified if the interview schedule applies to both the subjects and the therapist (Page 18, Lines 8-10). The examples of questions for each subscale should be reported consistently in either the text or brackets (Page 14, Lines 56-59). The abstract states that recruitment rates are evaluated as additional outcomes (Page 2, Lines 33-36), however, they are not thematized in the main text and should arguably be part of the primary outcomes. Lastly, it is not clear, why the qualitative outcomes (12.6) are not reported in the acceptability section (12.2), although acceptability will be rated largely based on the interviews (Page 12, Lines 45-47).

Authors> Thank you for this valuable feedback, the outcomes section has now been amended (pp. 12-20), the feasibility outcomes have now been described in more detail (p. 13), and an introductory paragraph has now been added to the physiological outcomes section (p. 16). Additionally, more details have now been added to the HRV methods (p. 16), exclusion of subjects with pain has now been clarified (p.17), and details of contacting in the event of an adverse event have been added to the Harms section (p. 25). The consistency issue with questions for each subscale has now been resolved (p15-17). The additional outcomes sentence in the abstract has now been removed, as the reviewer correctly points out these are part of the main feasibility and acceptability outcomes. Regarding the acceptability outcomes, we have moved the section to the qualitative methods in 12.2 as suggested.

Reviewer 1> Part of the recruitment process was described in the study setting section (Page 8, Lines 4-13), but may be more appropriately placed in the recruitment section (Page 19, Lines 43-57).

Authors> Thanks for this, this paragraph has now been moved to the recruitment section (#15, p. 22).

Reviewer 1> Harms that are common to osteopathic treatment may require a reference (Page 23, Lines 31-38). Further, it is unclear how the severity of adverse events will be defined (Page 23, Lines 50-52).

Authors> Thanks, a reference has now been added and the definition of severe adverse events has now been included.

47. Carnes D, Mars TS, Mullinger B, Froud R, Underwood M. Adverse events and manual therapy: A systematic review. *Manual Therapy*. 2010;15(4):355-63.

Reviewer 1> The consent section refers to supplementary material 2 (Page 25, Lines 12-15), which seems to be missing from the manuscript and submitted materials.

Authors> Thanks for bringing this to our attention, supplementary material 2 has now been included.

Reviewer 1> Information on competing interests and funding sources are outlined multiple times (Page 3, Lines 47-50; Page 4, Lines 12-15; Page 25, Lines 33-38; Page 27, Lines 3-8; Page 27, Lines 10-15), which appears redundant.

Authors> Thanks for this feedback, redundant information has now been removed where possible while still adhering to the SPIRIT and journal guidelines.

Reviewer: 2

Reviewer 2> The study is based on evidence that osteopathic interventions may also affect mental health symptomatology, e.g., by improving psychophysiological functioning indicated by HRV. Overall, the presented protocol (recruitment started in December 2022) makes a sound impression and I'm interested in the final study results. Please find some detailed comments below.

Authors> Many thanks to the reviewer for these kind comments.

Reviewer 2> In section 6.2 the references are used in a different style and some of them (e.g., Castro-Sanchez, 2014, Sherman et al., 2010) cannot be found in the references list. This makes it difficult to see which studies were actually referenced.

Authors> Thanks for raising this, the reference style has now been amended and citations are now organized correctly.

Reviewer 2> Most of the text is well written, however, there are some minor typos, missing commas and it makes sense to check carefully again (e.g., p. 11: The practitioner will first observe the participant while standing, then will observe active range of movements in with the participant in standing and/or sitting positions.). Furthermore, the description of psychophysiological literature could be a bit more differentiated as results concerning, e.g., HRV are not always unidirectional and interpretation often difficult. As feasibility of the interventions is the main concern of the study, this can be excused.

In section 7.0 Objective: "symptoms of mental health" should be rather "mental health symptomatology" or "symptoms of mental illness"?

Authors> Thank you for these suggestions. We have amended the sentence p. 10 which had a syntax error. It now reads as: "The practitioner will first observe the participant while standing, then will observe active range of movements with the participant in standing and/or sitting positions."

Regarding the HRV, we have now been more precise as to how the measurements will be taken, including the participant's positions (p. 18: "Participants will lie in a supine position while the ECG monitor records for at least 5 minutes").

p. 7, we have changed as suggested to "mental health symptomatology".

Reviewer 2> Why not use 32 participants, a number which can be divided by 4 groups?

The authors decided not to use a pure (time) control group, but to rather compare several potentially effective osteopathic interventions as the focus is on the feasibility of applying these to patients with mild and moderate mental health issues. However, are group differences expected or are all interventions thought to have the same effect?

Concerning section 12.4.1 HRV: It is not defined how long the measurement will be. Is there a baseline? Are participants seated? Are these measurements collected in the same environment as BP measurements? Furthermore, are participants asked to refrain from caffeine, alcohol, ... as these are factors that could potentially influence psychophysiological measurements?

Authors> Thank you, we agree that 32 participants is a much more logical sample size. We have amended the number in section 14, p. 22. The interventions are hypothesized to have a similar effect, but this has now been made clearer on p.8.

Regarding the HRV, the duration of the measurement (5 minutes) and supine position is now added (p.18). The environment will be the same for all physiological measures and this has now been made clear on p.17. Lastly, information has now been added about substances participants will be asked to refrain from (p. 18).

Reviewer 2> One of the measures used by the authors is the LF/HF quotient. Considering difficulty of interpretation (LF is thought to be influenced by sympathetic as well as parasympathetic ANS) of this measure (see e.g., Reyes del Paso, G. A., Langewitz, W., Mulder, L. J., Van Roon, A., & Duschek, S. (2013). The utility of low frequency heart rate variability as an index of sympathetic cardiac tone: a review with emphasis on a reanalysis of previous studies. *Psychophysiology*, 50(5), 477-487.) it may be a good idea to rethink HRV analyses.

Authors> Thanks for the suggested review paper, our main measure of HRV is the RMSSD. We will maintain the LF/HF measure but will interpret in line with the review paper you mention as a measure primarily of the parasympathetic system as this is still useful data on parasympathetic activity. We have now included the paper you mention and have recognised its potential limitations.

Reviewer: 3

Reviewer 3> Overall, the background and rationale section provides a clear introduction to the topic of mental health problems and the potential use of osteopathic interventions as a complementary approach. The section effectively highlights the increasing burden of mental health problems in the

UK and the limitations of traditional forms of care. The rationale for considering innovative approaches, such as osteopathy, is well justified.

Authors> We thank the reviewer for these supportive comments.

Reviewer 3> However, there are a few areas that could be improved or clarified: In line 27, the statement "All of the techniques used have an effect on the interplay between the nervous and musculoskeletal systems" could benefit from further elaboration or citation to support this claim. Providing specific examples or evidence for how osteopathic techniques influence the nervous and musculoskeletal systems would strengthen the argument.

Authors> Thank you for this comment, we have now added in references to support this statement on p. 5.

Reviewer 3> In line 58, the reference to a pre-registered systematic review on OSF is provided, but it would be beneficial to include a brief overview of the protocol and the expected outcomes or objectives of the review.

Authors> Thank you for this suggestion, a brief overview has now been added on p. 6.

Reviewer 3> By addressing these points, the background and rationale section would be further strengthened in terms of providing clear justification for the study and enhancing the credibility of the presented information.

Authors> Thank you for your feedback which has improved the background and rationale section.

Reviewer 3> Objectives. The objective is clear and concise, stating the purpose of the study to assess the feasibility and acceptability of four osteopathic interventions for adults with mild to moderate symptoms of mental health. The secondary aim is well-defined, indicating that the study also aims to assess the impact of the four interventions on physiological factors and their effectiveness in improving psychological outcomes. The hypothesis is straightforward and aligns with the objective of assessing feasibility and acceptability. The hypothesis is clear, stating that the interventions will lead to psychophysiological relaxation by increasing heart rate variability (HRV) and improving interoceptive accuracy. The final hypothesis is well-stated, indicating that the four interventions will result in improvements in measures of mental health.

Overall, the objectives and hypotheses are clearly articulated, providing a comprehensive understanding of the study's goals and expectations.

Trial design. The description of the trial design is clear, stating that an explanatory sequential mixed-methods approach will be used, with the quantitative aspect preceding the qualitative aspect. The description of the quantitative aspect of the trial design is concise and effective, indicating that a parallel, randomized design will be used with a 1:1 allocation to each of the four conditions. The qualitative aspect of the trial design is well-explained, mentioning that interviews will be conducted with both the participants of the intervention and the practitioners delivering them. The trial design section provides a clear overview of the approach, highlighting the use of both quantitative and qualitative methods in a sequential manner. The allocation method for the quantitative aspect is also specified.

Study setting. The study setting is clearly stated as Swansea University in South Wales, UK. The recruitment sources are also mentioned, including the student population and the general public, with specific methods such as advertising in communal spaces and utilizing social media. The information regarding the location and start date of the study is provided concisely.

Eligibility Criteria. The eligibility criteria are clearly defined, specifying the age range, target symptoms, and language proficiency required for participation. The rationale for excluding participants experiencing pain is provided, explaining the potential confounding effects on mental health outcomes.

Interventions. The description of the interventions is clear, stating the four types of osteopathic techniques and providing an estimated duration for each session. Referring to Table 1 for a summary adds clarity.

Modifications. The mention of potential modifications to ensure participants' safety and avoid discomfort is appropriate and responsible.

Adherence. The explanation regarding adherence is appropriate for a single-session intervention, and

the plan to record any instances of participants ending the session early adds transparency. Concomitant care. The plan to collect information about participants' concomitant care and account for it in the statistical analysis is appropriate and ensures that these factors are considered during the study.

Overall, the Methods section provides detailed information about the study setting, eligibility criteria, interventions, modifications, adherence, and concomitant care. The descriptions are clear and provide sufficient information for understanding the study design.

Outcomes. The section provides an overview of the outcomes that will be assessed in the study. It includes subsections on feasibility, acceptability, psychological outcomes, physiological outcomes, additional outcomes, and qualitative outcomes. The subsections provide clear descriptions of the measures and procedures that will be used to assess each outcome. The information is well-organized and easy to follow. Overall, this section provides a comprehensive overview of the outcomes that will be evaluated in the study.

Feasibility: The subsection describes the feasibility of the recruitment process and the measurement tools. It mentions that feasibility will be determined by the number of people who respond to advertisements and the number of people who are eligible/ineligible following the screening process. It also mentions that the feasibility of the measurement tools will be assessed based on participants' ability to complete the measures and the presence of missing data. The subsection provides a clear explanation of how feasibility will be evaluated. No specific issues were identified in this subsection.

Acceptability. The subsection provides a clear description of the methods for assessing acceptability. No specific issues were identified in this subsection.

Psychological outcomes. The subsection provides clear and concise information about the measures. No specific issues were identified in this subsection.

Physiological outcomes. The subsection provides clear information about the physiological measures. No specific issues were identified in this subsection.

Additional outcomes. The subsection provides a clear explanation of why these outcomes are being assessed and how they will be collected. No specific issues were identified in this subsection.

Qualitative outcomes. The subsection provides clear information about the qualitative data collection and analysis process. No specific issues were identified in this subsection.

Overall, the sections are well-written and provide a comprehensive overview of the outcomes that will be assessed in the study. The descriptions are clear and concise, making it easy to understand the measures and procedures that will be used. The subsections are logically organized and flow well. No major issues were identified in the sections reviewed.

Authors> We thank the reviewer for their supportive comments and for evaluating the manuscript so thoroughly against the SPIRT guidelines.

Reviewer 3> Sample Size. This section provides a clear and concise explanation of the sample size, but it would be helpful to mention what the previous research that informs the decision was and cite the sources.

Authors> Thank you for this suggestion, we have now added an additional citation with further justification for the sample size (p.20).

Reviewer 3> Recruitment. This section describes the recruitment process for participants, which involves contacting the research team, reading an information sheet, and completing the Depression, Anxiety, Stress Scale (DASS) to determine eligibility. The authors also mention that participants with severe mental health symptoms will be signposted to relevant mental health services and charities. The section is well-written and provides a clear and concise explanation of the recruitment process.

Authors>We thank you for the supportive comment.

Reviewer 3> Allocation. This section is clearly written and provides a detailed explanation of the randomization process, but it would be helpful to mention the sample size again and explain why four conditions were chosen.

Authors> Thank you, we added the sample size to the section: p. 23 "The 32 participants will be randomly (...)".

We decided not to justify the four conditions as this information is provided in section 11.1 "interventions".

Reviewer 3> Blinding (masking). This section explains how blinding will be ensured for the outcome assessor who will be blind to the participants' group allocation. The authors also state that the osteopathic practitioner will not be blinded to the intervention type but will be blinded to the study outcomes. The section is well-written and provides a clear explanation of the blinding process. Data management. This section is clear and concise and provides a detailed explanation of data management. Statistical Methods. This section briefly describes the statistical methods that will be used to analyze the data.

Authors> Thank you for the kind comments.

Reviewer 3> Overall, the sections of the paper reviewed are well-written, clearly explained, and provide a detailed description of the study design and methods. However, there are some areas where more information could be provided, such as the statistical methods used and why four conditions were chosen for the study. Additionally, citing the sources for the previous research that informed the sample size decision would strengthen the study's credibility.

Authors> Thank you very much for these suggestions which we have now incorporated into the manuscript.

VERSION 2 – REVIEW

REVIEWER	Bohlen, Lucas Osteopathie Schule Deutschland GmbH, Osteopathic Research Institute
REVIEW RETURNED	30-May-2023

GENERAL COMMENTS	I would like to thank the authors for the thorough revision of the manuscript. The changes implemented have sufficiently addressed my questions and suggestions. I have no further concerns to raise and recommend acceptance. I am looking forward to reading the results of this study.
---

REVIEWER	Keller, Micha RWTH Aachen University
REVIEW RETURNED	12-Jun-2023

GENERAL COMMENTS	Thanks for your appropriate responses to my own and the other reviewers' points of criticism. In my opinion all concerns have been addressed appropriately and I am looking forward to seeing the results soon.
---

REVIEWER	Esteves, Jorge Malta International College Osteopathic Medicine
REVIEW RETURNED	09-Jun-2023

GENERAL COMMENTS	Thank you for your work on the manuscript. It is a much improved version.
---